# Immunogenetic Role of IL17A Polymorphism in the Pathogenesis of Recurrent Miscarriage

**DOI:** 10.3390/jcm11247448

**Published:** 2022-12-15

**Authors:** Shafat Ali, Sabhiya Majid, Md. Niamat Ali, Mujeeb Zafar Banday, Shahnaz Taing, Saima Wani, Mansour Almuqbil, Sultan Alshehri, Kashif Shamim, Muneeb U. Rehman

**Affiliations:** 1Cytogenetics and Molecular Biology Laboratory, Centre of Research for Development, University of Kashmir, Srinagar 190006, J&K, India; 2Department of Biochemistry, Government Medical College, Srinagar 190010, J&K, India; 3Multidisciplinary Research Unit, Government Medical College, Srinagar 190010, J&K, India; 4Department of Obstetrics and Gynaecology, Government Medical College-Associated Lalla Ded Hospital, Srinagar 190008, J&K, India; 5Department of Obstetrics and Gynaecology, Sher-i-Kashmir Institute of Medical Sciences (SKIMS), Soura, Srinagar 190011, J&K, India; 6Department of Clinical Pharmacy, College of Pharmacy, King Saud University, Riyadh 11451, Saudi Arabia; 7Department of Pharmaceutics, College of Pharmacy, King Saud University, Riyadh 11451, Saudi Arabia; 8National Centre for Natural Products Research, University of Mississippi, Oxford, MS 38677, USA

**Keywords:** recurrent miscarriage, enzyme-linked immunosorbent assay, polymorphism, genotype, interleukin, restriction

## Abstract

Interleukin-17A (IL17A) is a proinflammatory cytokine and is assumed to play an important role in fetal rejection. In order to evaluate the potential role of IL17A polymorphism in the pathogenesis of recurrent miscarriage (RM), serum IL17A levels were estimated by ELISA. Single-nucleotide polymorphism was assessed by polymerase chain reaction-restriction fragment-length polymorphism (PCR-RFLP) using gene-specific primers and the EcoNI restriction enzyme. Serum IL17A levels were nonsignificantly (*p* > 0.5) low in RM patients compared with the control group. IL17A gene amplification by PCR yielded the undigested product of 815 bp, and its digestion with EcoNI enzyme produced 815, 529, 286, and 270 bp fragments for the GG genotype; 529, 286, and 270 bp fragments for the GA genotype; and 529 and 286 bp fragments for the AA genotype. The genotype frequency between the RM and control groups exhibited a significant difference (*p* = 0.001), whereas no significant difference was observed between allele frequencies in the two groups (*p* = 0.0954). These data suggest that the IL17A gene polymorphism exhibits no significant effect on IL17A gene expression. However, it significantly decreases and increases RM risk in the homozygous and recessive models, suggesting its potential pregnancy-protecting and -harming roles in the AA and GA + GG genotypes, respectively.

## 1. Introduction

Pregnancy is a tightly immune-regulated process with many immune cells activated, releasing an array of cytokines. Cytokines are considered responsible for embryonic tolerance or rejection. Proinflammatory cytokines tend to reject an embryo, while anti-inflammatory cytokines develop maternal immune tolerance for the embryo. The balance between the two types of cytokines may lead the pregnancy to full term; otherwise, the pregnancy may collapse midway [1,2]. The loss of two or more pregnancies is known as recurrent miscarriage (RM) or recurrent pregnancy loss (RPL) [3,4]. This reproductive disorder affects nearly 5% of couples worldwide [1,2], with known causes including endocrine defects, chromosomal defects, immune disorders, uterine defects, genital infections, and vitamin D3 deficiency [1,5].

Interleukin-17 (IL17) is a proinflammatory cytokine produced mainly by Th17 cells and takes part in immune inflammatory responses [6]. Various other cellular sources of IL17 include natural killer cells, monocytes, natural killer T cells, neutrophils, CD8^+^ T cells, eosinophils, γδT cells, innate lymphoid cells (specifically ILC3) [7,8,9,10,11], placental macrophages [12], and cytotrophoblasts and syncytiotrophoblasts from spontaneous pregnancy loss, normal term, and molar pregnancies [13]. The IL17 family consists of many closely linked proteins, including IL17A, IL17B, IL17C, IL17D, IL17E, and IL17F. The two most commonly studied IL17 family members are IL17A and IL17F, which exhibit higher similarity in protein sequence, receptor-binding, and biological functions [14,15]. IL17A and IL17F genes are present on chromosome 6p12 [16,17] (Figure 1). IL17A, IL17F, and other members of the IL17 family of cytokines bind to the IL17RA and IL17RC subunits of the IL17R (IL17 receptor) during IL17 signaling to transmit signals to target cells and produce effector molecules [18,19]. Both subunits (IL17RA and IL17RC) of IL17R are connected to the adaptor protein for recruiting TRAF6. The recruitment and subsequent ubiquitylation of TRAF6 with Act1 serving as an E3 ubiquitin ligase trigger a series of molecular interactions that cause the IκB phosphorylation and its subsequent degradation by proteasomes, enabling the NFκB translocation into the nucleus, where it targets genes encoding inflammatory cytokines and chemokines [19,20].

IL17A plays a pivotal role in host immunity and tissue inflammation [18]. IL17 increases host resistance against a wide range of infections caused by bacteria, fungi, viruses, and parasites [21,22,23] during pregnancy [24]. Abnormal IL-17A signaling has been linked to the development of various inflammatory and autoimmune disorders, including chronic respiratory diseases, idiopathic pulmonary fibrosis, acute lung damage, and cancer [25,26,27]. IL17 is essential in rejecting allograft [28,29,30,31,32,33,34], suggesting its potential role in dismissing the semiallogenic fetus. The intraperitoneal inoculation of recombinant IL17 induced fetal loss in a normal mouse model, whereas an anti-IL17 antibody prevented the fetal loss in an abortion-prone mouse model [35], suggesting the pivotal role of IL17 in spontaneous abortion in mice. IL17A exerts its proinflammatory effect via interactions with some mediators, including INFγ, IL1β, TNFα, IL22, and GM-CSF [7,26]. A healthy pregnancy exhibits a balance between Th17 and Treg cells, whereas an imbalance in Th17/Treg ratio with an increase in Th17 cells can be deleterious to pregnancy [36]. Th17 cells play a part in conceptus antigen rejection and could lead to obstetrical complications, including RM [30,37,38,39]. Several studies have reported an increased proportion of Th17 cells [30,40,41,42] and a decreased proportion of Treg cells in RM patients [40,41,42]. Consequently, higher expression of the IL17A gene has been reported in RM patients compared with healthy controls [43,44,45,46]. The risk of developing type 1 diabetes has been associated with higher serum IL17A levels [47]. It is believed that the IL17 activation may augment the expression of NFκB, which decreases the number of progesterone receptors and their function. As a result, progesterone binds inadequately to the fewer available progesterone receptors, resulting in decidual dysplasia, insufficient embryonic nourishment, and ultimately miscarriage [48].

Polymorphism in the IL17A gene may affect its expression in RM and many other diseases. Alkhuriji (2017) associated the IL17A SNP with RM [6]. Similarly, the IL17A SNP with the G allele has been found as a risk factor in RM women infected with Toxoplasma gondii [45]. However, Najafi et al., (2014) found no association between the IL17A (rs2275913) SNP and RM risk [49]. Some other studies have linked the IL17A gene variant with an increased risk of a disease, such as systemic lupus erythematosus [50], gastric cancer [51], and papillary thyroid carcinoma prognosis [52]. The role of IL17A in the current RM population has not been studied so far; therefore, this study was conducted to evaluate the association between IL17A polymorphism and the disease.

## 2. Materials and Methods

### 2.1. Study Design

This was a case-control outpatient clinic-based study.

### 2.2. Study Site

The study was carried out at outpatient antenatal clinics at the Department of Obstetrics and Gynecology, Government Medical College–associated Lalla Ded Hospital Srinagar, Jammu and Kashmir, India, from 20 June 2017 to 17 June 2022, where female patients, including RM patients, came across from Kashmir for a regular checkup. The lab work during this period was conducted at the Department of Biochemistry and Multidisciplinary Research Unit, Government Medical College, Srinagar, Jammu and Kashmir, India.

### 2.3. Study Participants

The study included the antenatal RM cases and controls (Figure 2) that fulfilled the below-given inclusion and exclusion criteria:

### 2.4. Inclusion Criteria for Patients

Pregnant women with a history of two or more successive miscarriages ≤24 weeks of gestation.Pregnant women with a history of primary RM (no history of childbirth before RM) or secondary RM (history of childbirth before RM).All the pregnant RM patients were from the female Kashmiri population.Pregnant RM patients of reproductive ages between 18 and 45 years were part of this study.Pregnant RM patients who willingly signed the consent form were included in the study.

### 2.5. Exclusion Criteria for Patients

Pregnant women with a history of only one miscarriage.Pregnant women with a history of two or more induced abortions and no spontaneous pregnancy losses.Nonpregnant women with a history of two or more spontaneous consecutive pregnancy losses.Pregnant women with a history of two or more spontaneous, nonconsecutive pregnancy losses.Pregnant women who conceived with ART/IVF.Pregnant RM patients who declined to participate in the study.

### 2.6. Inclusion Criteria for Controls

Pregnant women with a history of one or more successful pregnancies.Pregnant women who belonged to the Kashmiri population.Pregnant women of ages between 18 and 45 years were included in the study.

### 2.7. Exclusion Criteria for Controls

Pregnant women with no birth history (i.e., women carrying their first pregnancy).Women with no history of pregnancy.Pregnant women with a history of miscarriages.Pregnant women who conceived with ART/IVF.Pregnant women who declined to participate in the study.

### 2.8. Ethical Approval and Patient Consent

This study was approved by the Institutional Ethical Committee, Government Medical College, Srinagar, Jammu and Kashmir, India, with reference No. 121/ETH/GMC. Informed consent in both English and vernacular was taken from the participants.

### 2.9. Collection of Samples

Peripheral blood (PB) samples (4–5 mL) were drawn from each participant and equally distributed into blood clotting activators and ethylenediaminetetraacetic acid (EDTA) vials. The blood clotting activator vials were left undisturbed at room temperature for some time. After that, the vials were centrifuged at 4000 rpm for 5 minutes, and the serum was collected and transferred to eppendorf tubes. The serum samples were stored at −20 °C until analyzed by enzyme-linked immunosorbent assay (ELISA). The EDTA vials were stored at −80 °C until molecular studies on IL17A single-nucleotide polymorphism by restriction fragment-length polymorphism-polymerase chain reaction (RFLP-PCR) was carried out. Whole deoxyribonucleic acid (DNA) was extracted by the GenElute^TM^ Blood Genomic DNA Kit (Sigma-Aldrich, St. Louis, USA) with some modifications.

### 2.10. Estimation of Serum IL17A Cytokine Levels

IL17A cytokine was measured by the Sandwich Human IL17A ELISA kit (Diaclone, Cat. No. 850940096, Besancon, France) following the manufacturer’s instructions (Figure 3). Standards (serially diluted), diluted samples (patient and control), and zero (standard diluent buffer 1X) 100 μL each were added to their corresponding wells. The wells were covered with a plastic cover and incubated for 2 h at room temperature (18–25 °C), followed by washing each well with 300 μL of 1X wash buffer 3 times in an automatic washer (LabLife, Germany). Subsequently, diluted biotinylated antibody (50 μL) was added to each well, and the plate was incubated for 1 h at room temperature, followed by washing (similar to the first washing). After washing, diluted Streptavidin-HRP (100 μL) was dispensed into each well, and again the plate was incubated at room temperature for 30 minutes and subsequently washed 3 times (similar to the first washing). TMB substrate (100 μL) was added to each well, and the plate was incubated in the dark for 5–15 minutes to develop color. The substrate reaction was stopped by adding H_2_SO_4_ stop reagent (100 μL) to each well. The absorbance value (OD) of each well was determined at the wavelength of 450 nm in a Tri-Star multimode reader (Berthold, Germany) with ICE software, immediately after adding the stop solution.

### 2.11. Extraction of Whole Genomic DNA

Whole DNA was extracted from blood samples with the DNA extraction kit (Sigma-Aldrich, St. Louis, MO, USA) following the manufacturer’s instructions to obtain purified and high-quality genomic DNA. The extracted DNA was subjected to agarose gel electrophoresis (Tanon, China), and the DNA bands obtained (Figure 4) were observed by the gel-documentation system (Aplegen, Pleasanton, CA, USA). Moreover, the concentration and the purity were measured by Nanodrop2000 (Thermo Scientific, Waltham, MA, USA). The elute containing pure genomic DNA was stored at −80 °C until further analysis.

### 2.12. Polymerase Chain Reaction (PCR)

The DNA fragment of interest from the IL17A gene was amplified by PCR using specific primers: Forward primer: 5′-TCT CCA TCT CCA TCA CCT TTG-3′ and reverse primer: 5′-GTC CAA ATC AGC AAG AGC ATC-3′ under the following specific PCR conditions: initial denaturation (94 °C/5 min), denaturation (94 °C/59 s), annealing (57 °C/59 s), extension (72 °C/59 s), and final extension (72 °C/10 min). The PCR reaction mixture (25 μL) consisted of buffer with MgCl_2_ (concentration/volume; 10X/2.7 μL); MgCl_2_ (20 μM/0.2 μL, added additionally); dNTP mixture (10 μM/2 μL; 0.5 each); gene-specific primers, specifically forward (10 μM/0.7 μL) and reverse (10 μM/0.7 μL); Taq polymerase (2 Units/0.4 μL); nuclease-free water (15.3 μL); and DNA template (50 ng/μL/2 μL). The genes were amplified on Thermocycler (Applied Biosystems, Life Technologies, Model #: 9902, Made in Singapore). The PCR amplicon of the IL17A gene was run on a 2% agarose gel and subsequently visualized in the gel-documentation system. The buffer, MgCl_2_, dNTPs, and Taq polymerase were procured from Sigma-Aldrich, St. Louis, MO, USA, however, primers were purchased from Merck, Germany.

### 2.13. Restriction Fragment-Length Polymorphism (RFLP)

The PCR amplicon of the IL17A gene was digested using EcoNI restriction enzyme (BioLabs, Cambridge, MA, USA). The genotyping information and protocol for studying IL17A SNP is described in Table 1. The bands obtained were separated by electrophoresis on a 2.5% agarose gel. The DNA bands were observed in the gel-documentation system. The representative PCR samples of IL17A were sent for sequencing to Biologia Research India Pvt. Ltd., Delhi, India. The samples were purified from agarose gel and sequenced by using the Sanger method.

### 2.14. Statistical Analysis

The statistical analysis of the collected data was carried out in SPSS (Version 22.0, Chicago, IL, USA), GraphPad Prism (Version 8.0), and Microsoft Excel (Version 12.0). Continuous variables were summed up by using number (N), standard deviation (SD), and range (min-max). On the other hand, categorical variables were summarized by percentages and frequencies. Chi-square (χ^2^) and *t* tests were employed to compare groups. Allele and genotype frequency differences between the cases and control groups were tested for significance by using Fisher’s exact test. Conditional logistic regression analysis was used to determine odds ratios (ORs) and 95% confidence intervals (CIs) to evaluate the potential association of the relevant SNP genotypes with RM, and to assess the possible gene-environment interaction. The magnitude of the effect was estimated by using odds ratios and their 95% confidence intervals. Hardy Weinberg equilibrium (HWE) fitness of the genotype distributions for the allele and genotype frequencies in the study population was tested using the Chi-square test. All the *p*-values presented were two-tailed. Results were assumed to be statistically significant at *p* < 0.05.

## 3. Results

### 3.1. Analysis of Serum IL17A Cytokine Estimation by ELISA

The serum levels of IL17A cytokine in both RM patients and the control group were quantified with the Human IL17A ELISA kit following the manufacturer’s guidelines. A standard linear curve was generated by plotting the OD of each standard on the y-axis and their corresponding known concentrations on the x-axis in Microsoft Excel (version 12.0). The unknown concentration of each sample was measured by extrapolating their OD values against the known standard concentrations of IL17A using the standard curve. The average levels of IL17A in both groups are demonstrated in Table 2. The average concentration of IL17A decreased in RM patients compared with the control group. The color presentation of the IL17A concentration of each sample was demonstrated in the heat map plot (Figure 5). However, the difference between the levels of IL17A in the RM and control groups was not statistically significant (*p* > 0.5) (Figure 6). The diagnostic potential of IL17A for this disease was assessed by the plotting ROC (receiver-operating characteristic) curve (Figure 7).

### 3.2. Analysis of IL17A Genotyping (PCR-RFLP)

PCR amplification of the IL17A gene yielded an undigested product of 815 bp (Figure 8). The sequencing pattern of the IL17A PCR product is shown in Figure 9. The restriction digestion of IL17A PCR amplicon with EcoNI enzyme produced 815, 529, 286, and 270 bp fragments for the GG genotype (Homozygous Wild), 529, 286 and 270 bp fragments for the GA genotype (Heterozygous), and 529 and 286 bp fragments for the AA genotype (homozygous variant) (Figure 10). In IL17A polymorphism, the distribution frequencies for the GG, AG, and AA genotypes in RM and the control groups were 51.5%, 29.9%, and 18.6% and 51.5%, 45.4%, and 3.09%, respectively. The allele frequencies for G observed in RM and the control groups were 66.5% and 74.2%, respectively, while the A allele frequencies observed in RM and the control groups were 33.5% and 25.8%, respectively. The genotype frequency between RM and the control groups exhibited a significant difference (*p* = 0.001), whereas no significant difference was observed between allele frequencies in the two groups (*p* = 0.0954). The potential role of IL17A in the risk of RM was evaluated by using five genetic models: GG vs. GA; AA vs. GA + GG; GG vs. AA; GG vs. GA + AA; and G vs. A (Table 3). IL17A gene polymorphism was found nonsignificantly associated with increased RM risk in the heterozygous model (GG vs. GA; OR: 1.517; 95% CI: 0.814–2.755; *p* = 0.2169) and significantly in the recessive model (AA vs. GA + GG; OR: 7.139; 95% CI: 2.177–23.42; *p* = **0.0008**). On the other hand, the homozygous model (GG vs. AA; OR: 0.166; 95% CI: 0.050–0.583; *p* = **0.0031**) exhibited a significant role in decreasing RM risk, while the dominant model (GG vs. GA + AA; OR: 1.00; 95% CI: 0.559–1.787; *p* > 0.9999) seemed to have no role in RM risk. However, the codominant/allelic model (G vs. A; OR: 0.689; 95% CI: 0.446–1.077; *p* = 0.1194) was found to be nonsignificantly involved in decreasing RM risk. 

The overall association of the IL17A SNP with RM was found to be significant (*p* = **0.0010**). In addition, the combined variant genotype (GA + AA) was found to have an overall significant association with RM (*p* = **0.0032**). Furthermore, the overall association of combined genotype (GA + GG) with RM was found to be significant (*p* = **0.0055**).

Further, the IL17A genotype frequencies were in agreement with HWE (χ^2^ = 3.34; *p* = 0.1883) among the controls but deviated among the patients (χ^2^ = 10.50; *p* = **0.0052**). 

## 4. Discussion

Proinflammatory cytokines are usually considered harmful to pregnancy. IL17 is a proinflammatory cytokine that plays an important role in immunoinflammatory responses [6]. Our study found decreased IL17A levels in RM patients compared with the control group. The difference in IL17A levels between the patient and control groups, however, was statistically insignificant (*p* > 0.05). These results suggest that IL17A levels remain constant during pregnancy, as the IL17A gene expression between the two groups exhibited a nonsignificant difference. Thus, peripheral IL17A seems to have no deleterious effects on pregnancy.

However, the results of most of the studies on serum IL17 levels were contrary to our results. IL17-producing cells show higher prevalence in the decidua and the PB of idiopathic RM women [30,53]. IL17 is more highly expressed in idiopathic RM patients than in nonpregnant healthy women [54]. Significantly higher IL17 levels have been reported in RM patients than in controls [45,55]. Sereshki et al. (2014) also reported augmented IL17 levels in the proliferative and secretory phases of RM patients [41]. Cai et al. (2016) reported increased IL17 expression in the decidua and PB of idiopathic RM women [56]. Several studies found that RM patients exhibit a decline in Tregs and an elevation of IL17-generating cells in their PB and decidua [57,58]. Serum IL17 increases in women with healthy pregnancies toward the end of pregnancy but decreases after pregnancy loss, suggesting the important regulatory role of IL17 in the successful completion of pregnancy [59]. IL17 activates different mediators of inflammation. Both peripheral and decidual Th17 cells increase in idiopathic RM women [30,54]. Previous studies on Th17 cells focused on inflammation, the dismissal of allografts, and autoimmune disorders, including inflammatory bowel disease [60,61]. Several studies have been conducted in recent years that investigated the involvement of Th17 cells in RM. Th17 cells are a new CD4+ T cell subpopulation that can be useful in pregnancy tolerance. Nakashima et al. studied the Th17 cell proportion in PB during the first, second, and third trimesters of pregnancy and reported that the Th17 cell remains consistent during pregnancy [62]. In our study, the comparable serum IL17A levels between the RM and control groups suggest that the proportion of Th17 cells remains constant during pregnancy in both RM patients and controls. According to Lee et al., there is an imbalance between Th1 and Th2 cells. They hypothesized that higher Th17 cell counts and a better Th1/Treg ratio would trigger an inflammatory response that would ultimately aid RM development. In addition, they found an increased IL17+ T cell proportion in the PB of nonpregnant women with RM history. It is thought that these cells play a significant role in the inflammatory immunological responses that occur at the maternoembryonic interface during implantation and may later result in RM development [63].

RM has also been associated with chronic endometritis [64,65]. It is a unique endometrial inflammatory disorder that is very common among women with reproductive failure [66]. Chronic endometritis is believed to impair endometrial receptivity [67,68,69,70] via activating local inflammatory processes, leading to altered cytokine and chemokine secretion [69,71,72,73,74]. Many studies have reported higher Th17 cells and IL17 in the peritoneal fluid [75,76] and serum [77] of patients with endometritis and chronic endometriosis [69]. Thus, a higher expression of IL17 seems to be involved in the pathogenesis of endometriosis and chronic endometriosis that may cause reproductive failure, including RM.

In this study, we also assessed the diagnostic potential of IL17A for this disease by plotting the ROC curve. However, it was found that IL17A had no discriminatory diagnostic potential for RM on the basis of the AUC (area under the ROC curve) (area (CI 95%); *p*-value = 0.584 (0.502–0.666); 0.0424). The overall diagnostic precision of the test may be effectively summarized using AUC. AUC measures discrimination and allows the diagnostic performances of two tests to be compared [78]. It accepts values ranging from 0 to 1, 0 indicating a perfectly incorrect test and 1 indicating a perfectly accurate test [78]. Generally speaking, an AUC of 0.5 indicates no discrimination (i.e., the ability to diagnose individuals with and without the disease or condition based on the test), 0.7 to 0.8 is regarded as satisfactory, 0.8 to 0.9 is considered excellent, and more than 0.9 is considered remarkable [79]. In our study, the AUC obtained for IL17A was 0.584. Therefore, IL17A cannot be used as a significant diagnostic marker for RM, because its distributional difference between the patient and control groups is insignificant.

The study investigated the possible connection between IL17A polymorphism and RM risk and found no significant association between IL17A polymorphism and RM risk in most of the genotypes (heterozygous, dominant, and allelic). Our results were supported by many studies that found no statistically significant association between IL17 polymorphism and RM risk [49,80,81]. We found a significant (*p* = **0.0008**) risk of RM associated with the GA + GG genotype. This finding was supported by several other studies that reported a significant association between the IL17A SNP and RM risk [6,45,82]. IL17 gene variants have also been associated with higher miscarriage risk [80,83]. In our study, however, IL17A polymorphism appeared to reduce RM in the AA genotype.

Our findings revealed that the genotype frequencies of the IL17A polymorphism showed a statistically significant difference between the RM and control groups. The GG genotype exhibited the highest frequency, with an equal frequency in both the RM and control groups (51.5%). The G allele was also highest in both groups (66.5% vs. 74.2%). However, Najafi et al. (2014) reported the AA genotype with the highest frequency in the RM and control groups (61.2% vs. 54.1%). Allele A also exhibited a higher frequency distribution in both of the groups (76.5% vs. 75.3%) [49]. Our study also found higher AA genotype frequency in the RM group compared to control group (18.6% vs. 3.09%). 

Additionally, IL17A polymorphism has been assessed in many other diseases in diverse populations. The different studies on the IL17A polymorphism have shown contradictory findings. Wang et al. investigated IL17A polymorphisms in Chinese Han breast cancer patients. According to their findings, the patient group and the control group exhibited AG genotype frequencies of 47.66% and 48.9%, respectively. Moreover, they reported the association of IL17A SNPs with the risk of breast cancer [16]. Quan et al. investigated the link between IL17 gene variants and cervical cancer risk in Chinese women. They reported the AG frequencies of 45.7% and 46.4% in women with cervical cancer and healthy controls for IL17A SNPs, respectively. Furthermore, the IL17A SNP (rs2275913) was significantly associated with the risk of developing cervical cancer [84]. The IL17A SNP has been reported as a risk factor for developing hepatitis B virus-associated hepatocellular carcinoma in the Chinese Han population [85].

IL17A polymorphism has also been reported as a risk factor for lupus nephritis [41]. Moreover, the IL17A gene variant has been linked to an increased risk of colorectal cancer in Asians and Caucasians [86,87], osteoarthritis in Caucasians [88], IgA nephropathy in Chinese Hans [89], polycystic ovarian syndrome in Chinese people [90], diabetic nephropathy in Iranians [91], diverse cancer in Asians [92], preeclampsia in Chinese people [93], and rheumatoid arthritis in Europeans [94]. However, IL17A polymorphism has been reported to show no association with many diseases, such as European and Asian psoriasis [95], Egyptian multiple myeloma [96], and Polish periodontitis [97].

In our study, the overall association of IL17A SNP with RM was found to be significant (*p* = **0.001**). Most of the genotypes seem to have no effect on IL17A gene expression. The AA genotype seems to be pregnancy protecting and may regulate the IL17A gene expression to levels safe for pregnancy, whereas the GA + GG genotype is pregnancy harming that may overactivate the IL17A gene expression to levels harmful for pregnancy. However, the average serum IL17A levels remained the same between RM patients and controls which suggests that the IL17A polymorphism exerted no significant effect on IL17A gene expression. 

There is a further need for comprehensive studies on the association between RM and IL17A polymorphism as it seems to behave differently in different populations and different health conditions, which may be correlated with differences in geographical distribution, food habits, environmental status, familial history, ethnicity, and health status among populations. Moreover, in the cytokine microenvironment, the possible involvement of other proinflammatory and anti-inflammatory factors may affect the production and activity of IL17A.

## 5. Conclusions

IL17A levels were comparable in the patient and control groups. The IL17A gene polymorphism exhibited no significant effect on IL17A gene expression. In most genotypes, there was no significant association between IL17A polymorphism and RM risk. However, IL17A polymorphism significantly decreased the risk of RM in the homozygous model, suggesting its potential pregnancy-protecting role in the AA genotype. At the same time, it significantly increased RM risk in the recessive model, suggesting its deleterious role in pregnancy in the GA + GG genotype. The overall association of the IL17A gene polymorphism with RM was significant. Moreover, serum IL17A demonstrated no discriminatory diagnostic potential for RM. More exhaustive studies on IL17A together with other proinflammatory and anti-inflammatory factors may help shed better light on the role of IL17A in RM.

## Figures and Tables

**Figure 1 jcm-11-07448-f001:**
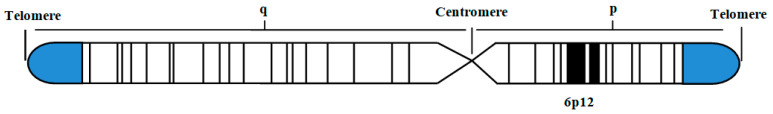
Location of the IL17A gene on the short arm (p) of chromosome 6.

**Figure 2 jcm-11-07448-f002:**
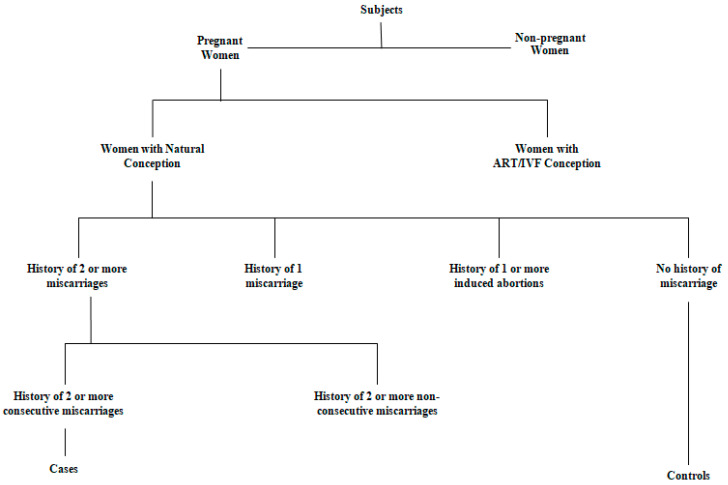
A schematic presentation for the recruitment of case-control subjects.

**Figure 3 jcm-11-07448-f003:**
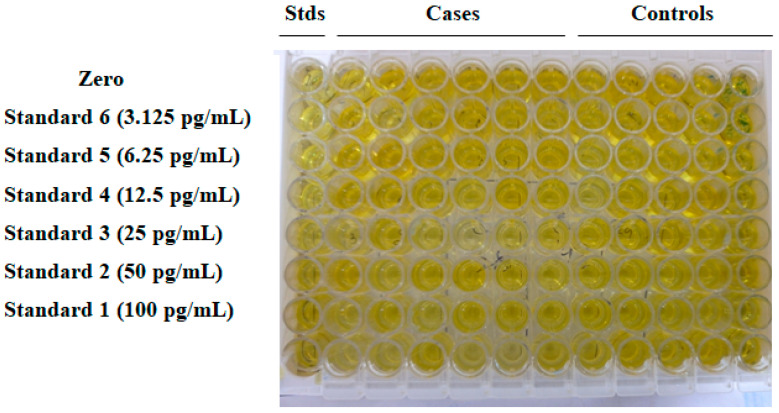
ELISA microtiter wells of IL17A. Standard diluent buffer 1X was taken as zero.

**Figure 4 jcm-11-07448-f004:**
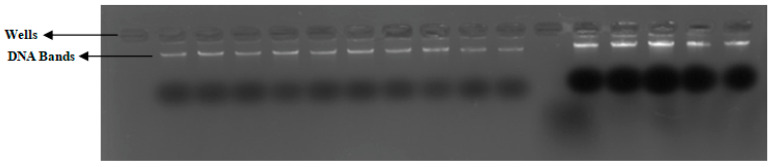
Pattern of DNA (whole DNA extracted from PBMN cells) bands obtained on 2% agarose gel.

**Figure 5 jcm-11-07448-f005:**
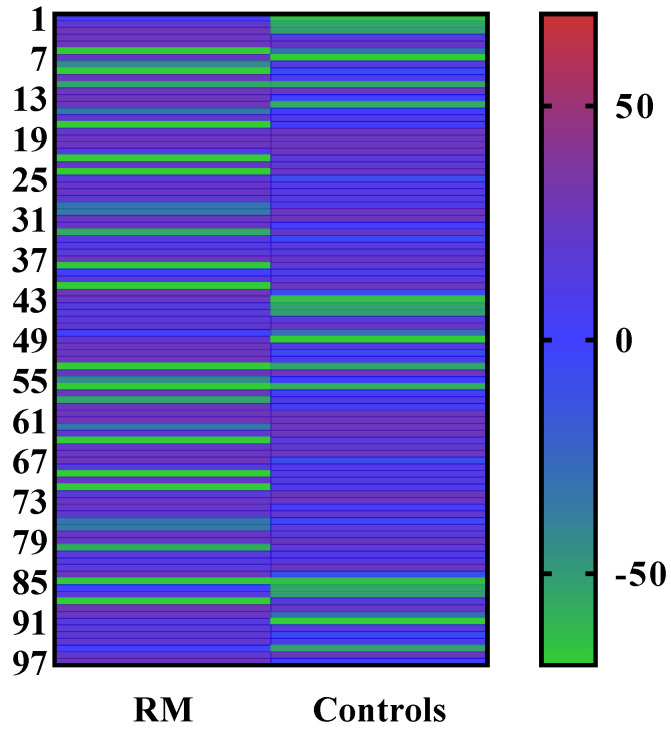
Heat map plot of IL17A samples (case vs. control).

**Figure 6 jcm-11-07448-f006:**
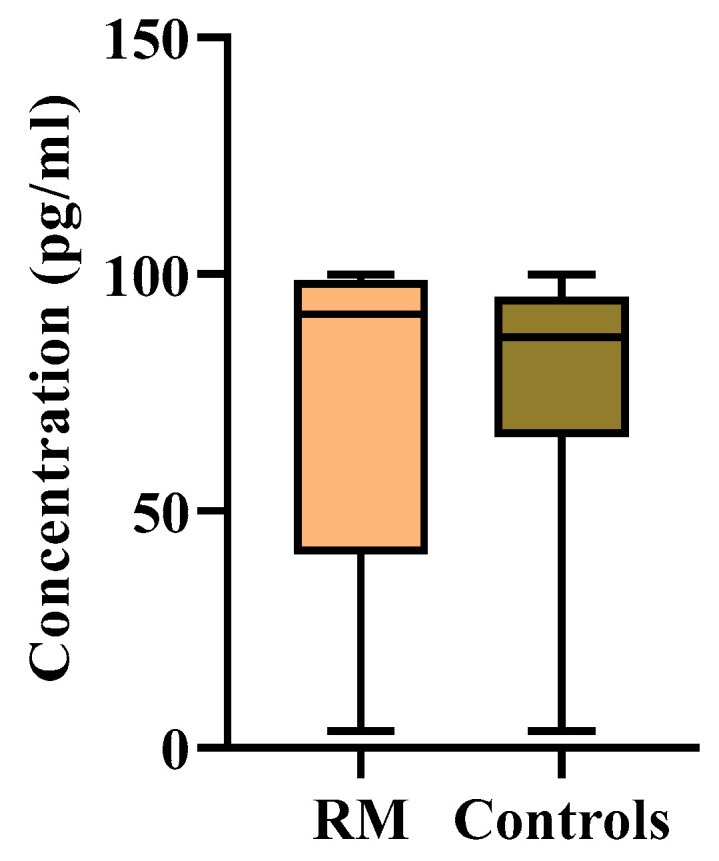
Box and Whisker plot showing serum IL17A concentration (case vs. control).

**Figure 7 jcm-11-07448-f007:**
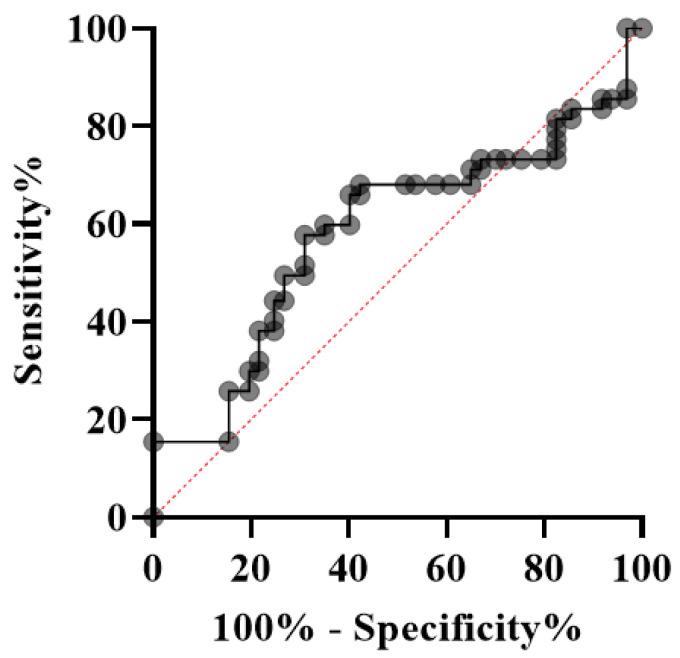
ROC curve of IL17A. Area (CI 95%); *p*-value = 0.5843 (0.5020–0.6667); 0.0424. The curve illustrates the diagnostic potential of IL17A for this disease.

**Figure 8 jcm-11-07448-f008:**
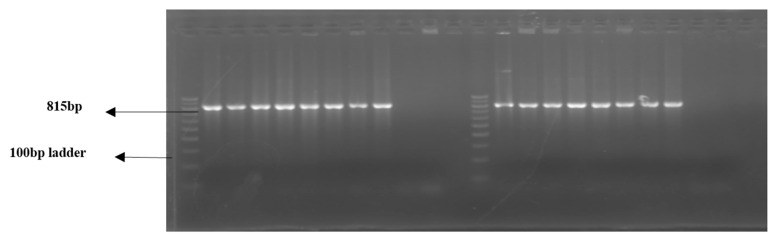
Pattern of PCR amplification product of the specific IL17A gene fragment (815 bp).

**Figure 9 jcm-11-07448-f009:**
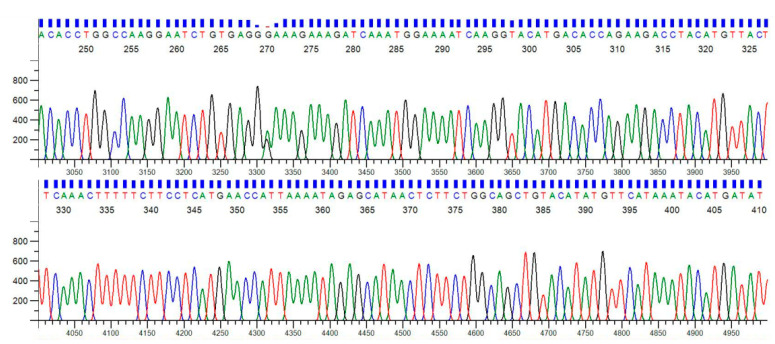
Forward pattern of a representative partial chromatogram of IL17A where each nitrogenous base is represented by a specific colour: adenine (green), guanine (black), thymine (red), and cytosine (blue).

**Figure 10 jcm-11-07448-f010:**
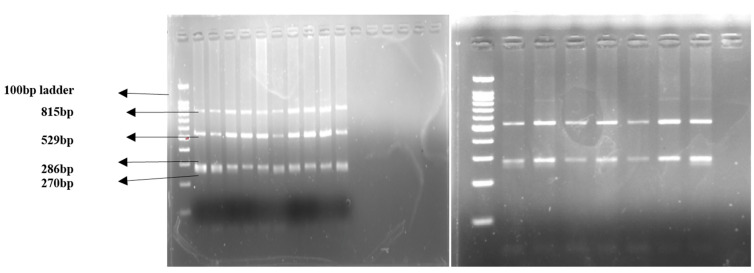
Pattern of IL17A restriction products after digestion with EcoNI enzyme.

**Table 1 jcm-11-07448-t001:** Genotyping information and protocol for studying the IL17A SNP.

Buffer	2 μL
Restriction enzyme (FastDigestion, BioLabs)	EcoNI (1 μL)
Nuclease-free H_2_O	7 μL
PCR product	6 μL
Incubation temperature (Drybath, Thermo Scientific, Korea)	37 °C
Incubation duration	1 h
PCR product size	815 bp
Restriction enzyme cut site	CCTNN/NNNAGG
Reference SNP cluster ID	rs2275913
Restriction digestion products	815, 529, 286, and 270 bp fragments

**Table 2 jcm-11-07448-t002:** IL17A serum concentration (case vs. control).

Cytokine		N	Mean	Mean ± SD	SE	Cl (95%)	*p*-Value
IL17A	Case	97	73.06	73.06.9 ± 35.8	3.6	7.2	0.9741
	Control	97	73.21	73.21 ± 29.9	3.04	6.0	

Two-tailed *t* test; *p* ≤ 0.05 is taken as significant.

**Table 3 jcm-11-07448-t003:** Pattern of genotype and allele frequencies in IL17A (G/A) gene polymorphism.

	Genotype	RM Patients	Controls	OR (95% CI); *p*-Value	RR (95% CI)	χ^2^; Pearson’s *p*-Value (Overall)
		(N = 97)	(N = 97)			
**Models Inherited**	GG	50 51.50%	50 51.50%	**1.0 (Reference)**		
**GG vs. GA** **(heterozygous)**	GA	29 29.90%	44 45.40%	1.517 (0.814–2.755); 0.2169	1.259 (0.907–1.795)	13.80; **0.0010** **13.80; 0.0032** **14.62; 0.0055**
**GG vs. AA** **(homozygous)**	AA	18 18.60%	3 3.09%	0.166 (0.050–0.584); **0.0031**	0.583 (0.454–0.800)	
**GG vs. GA + AA** **(dominant)**	GA + AA	47 48.50%	47 48.50%	1.00 (0.559–1.787); >0.9999	1.00 (0.754–1.329)	
**AA vs. GA + GG** **(recessive)**	GA + GG	79 81.40%	94 96.90%	7.139 (2.177–23.42); **0.0008**	1.877 (1.385–2.326)	
	**Allele**					
**G vs. A** **(allelic/codominant)**	G	129 66.50%	144 74.20%	**1.0 (Reference)** 0.689 (0.446–1.077); 0.119	0.836 (0.686–1.033)	2.781: **0.0954**
	A	65 33.50%	50 25.80%			

ORs (95% CI) were calculated with conditional logistic regression models; *p*-values were calculated with Chi-square tests; N (Number of individuals); OR (Odds ratio); CI (Confidence interval); RR (Relative risk); χ^2^ (Chi-square).

## Data Availability

The data presented in this study are available on request from the corresponding author.

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
