# Peer review of "Immunogenetic Role of IL17A Polymorphism in the Pathogenesis of Recurrent Miscarriage"

_jcm, 2022, doi:10.3390/jcm11247448_

Round 1
Reviewer 1 Report
Manuscript ID: #jcm-2046934
Title: " Immunogenetic role of IL17A polymorphism in the pathogenesis of recurrent miscarriage”
Comments for the author
I enjoyed reading this manuscript. It is well written and focuses on an important subject.
I have only a minor comment regarding typo errors that should be corrected. Maybe it would be a good option to send this manuscript to a professional proof reading English service.
Author Response
Reply to comments
Thanks for your comments. The manuscript has improved a lot. All the suggested changes are incorporated and queries are addressed in the revised manuscript. All the changes are highlighted.
Reviewer 1
I enjoyed reading this manuscript. It is well written and focuses on an important subject.
I have only a minor comment regarding typo errors that should be corrected. Maybe it would be a good option to send this manuscript to a professional proofreading English service.
Response: The whole manuscript has been proofread and all the mistakes and typo errors have been corrected in the revised manuscript.
Reviewer 2 Report
Introduction:
-line 58-61, please add a reference.
-line 61-62 please add definitions of RM and RPL, following international society guidelines.
Material and Methods:
- specify the years of the study, since "last five years" is valid only for the 2022.
-why did you not consider the infertile (IVF center related) population as a possible confounding factor?
Discussion:
- please consider also the possible chronic endometritis relation with IL17.
- please correct some typos and mistakes.
NB: Is the figure original? It seems like a scanned one from a book or chapter.
NB: Please increase the quality of figure 2, 3 and 4 (especially the text).
Author Response
Reply to comments
Thanks for your comments. The manuscript has improved a lot. All the suggested changes are incorporated and queries are addressed in the revised manuscript. All the changes are highlighted.
Reviewer 2
Introduction:
-line 58-61, please add a reference.
Response: References were already added but are now cited as per your suggestions
- Ali, S.; Majid, S.; Ali, M. N.; Taing, S. Evaluation of T cell cytokines and their role in recurrent miscarriage. Immunopharmacol. 2020, 82, 106347.
- Ali, S.; Majid, S.; Ali, M. N.; Taing, S.; Rehman, M. U.; Arafah, A. Cytokine imbalance at materno-embryonic interface as a potential immune mechanism for recurrent pregnancy loss. Immunopharmacol. 2021, 90, 107118.
-line 61-62 please add definitions of RM and RPL, following international society guidelines.
Response: RM/RPL is redefined according to the following international society guidelines
- ESHRE Guideline Group on RPL, Bender Atik, R., Christiansen, O. B.; Elson, J.; Kolte, A. M.; Lewis, S.; Middeldorp S.; Nelen, W.; Peramo, B.; Quenby, S.; Vermeulen, N.; Goddijn, M. ESHRE guideline: recurrent pregnancy loss. Reprod. open., 2018, 2018, 2, hoy004.
- Practice Committee of the American Society for Reproductive Medicine. Definitions of infertility and recurrent pregnancy loss: a committee opinion. Steril., 2020, 113, 3, 533-535.
Material and Methods:
- specify the years of the study, since "last five years" is valid only for the 2022.
Response: The study was carried out from 20-06-2017 to 17-06-2022. The dates are added in the revised manuscript.
-why did you not consider the infertile (IVF center related) population as a possible confounding factor?
Response: The aim of our study was to study the role of IL17 in recurrent miscarriage patients (natural pregnancy and spontaneous pregnancy losses) vs. controls with natural pregnancies. ART/IVF pregnancies and induced abortions were excluded from the study. However, IVF vs. recurrent miscarriage could be an interesting topic and may open a new research avenue in reproductive biology.
Discussion:
- please consider also the possible chronic endometritis relation with IL17.
Response: A short paragraph on the possible chronic endometritis relation with IL17 and recurrent miscarriage has been added to the discussion.
- please correct some typos and mistakes.
Response: Proofreading of the manuscript was done and all mistakes and typo errors have been corrected in the revised manuscript.
NB: Is the figure original? It seems like a scanned one from a book or chapter.
Response: The new original figure of the chromosome is added in the revised manuscript.
NB: Please increase the quality of figure 2, 3 and 4 (especially the text).
Response: The revised figures 2, 3 & 4 are added in the revised manuscript.